# GeoVLM-R1: Reinforcement Fine-Tuning for Improved Remote Sensing Reasoning

## Abstract

Recent advances in reinforcement learning (RL) have delivered strong reasoning capabilities in natural image domains, yet their potential for Earth Observation (EO) remains largely unexplored. EO tasks introduce unique challenges, spanning referred object detection, image/region captioning, change detection, grounding, and temporal analysis, that demand task-aware reasoning. We propose a novel post-training framework that incorporates task-aware rewards to enable effective adaptation of reasoning-based RL models to diverse EO tasks. This training strategy enhances reasoning capabilities for remote-sensing images, stabilizes optimization, and improves robustness. Extensive experiments across multiple EO benchmarks show consistent performance gains over state-of-the-art generic and specialized vision–language models. Code and models will be released publicly.

## 1 Introduction

Recent advances in remote sensing vision–language models (RS-VLMs) show strong performance on high-resolution Earth Observation (EO) imagery (Hu et al., 2023; Soni et al., 2025; Irvin et al., 2024; Zhan et al., 2024). However, these gains come with shallow reasoning: models rely heavily on text priors (Bleeker et al., 2024) and supervised finetuning (SFT) without chain-of-thought reasoning, leading to poor generalization. Early attempts with Reinforcement Learning (RL) as a post-training mechanism, such as UAV-VL-R1 (Guan et al., 2025), remain confined to visual question-answering (VQA) tasks only and perform poorly on broader EO tasks like detection, captioning, grounding, or disaster assessment (Soni et al., 2025). While RL offers the promise of reward-driven reasoning, existing approaches in EO receive weak and task-agnostic reward signals, making them vulnerable to reward hacking (Fu et al., 2025) and unable to capture the structured, multi-step reasoning demanded by complex EO scenarios (Li et al., 2025). A key challenge is thus building EO-VLMs that can reason robustly across complex and diverse tasks.

To address these challenges, we introduce GeoVLM-R1, a RL framework that enhances geospatial VLM reasoning while remaining flexible, scalable, and easy to extend across diverse EO tasks. To this end, our approach builds on group relative policy optimization (GRPO) (Shao et al., 2024) rather than standard proximal policy optimization (PPO) (Schulman et al., 2017) or direct preference optimization (DPO) (Rafailov et al., 2023), leveraging group-wise relative advantages to reduce training variance and improve structured reasoning. Central to GeoVLM-R1 is a dual-objective reward design: (i) accuracy compliance, ensuring semantic correctness, and (ii) format compliance, enforcing interpretable, structured outputs. Specifically, we introduce a task-aware accuracy reward mechanism that is designed to select a specific reward for each downstream EO task. For instance, in grounding-description

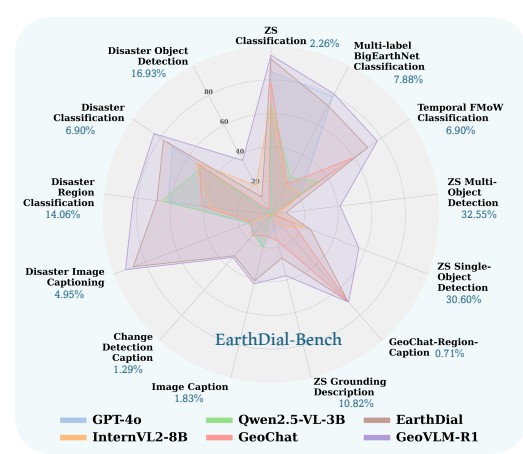

Figure 1: Comparison of recent generic and specialized VLMs over diverse EO tasks. GeoVLM-R1 shows favorable improvements across classification, detection, and captioning tasks.

tasks that require both object detection and textual explanation, simple similarity matching is insufficient; instead, we integrate bounding-box IoU with semantic alignment to jointly reward based on detection and description quality. Analogous task-specific rewards are defined for classification, change detection, captioning, and disaster assessment, ensuring targeted skill acquisition without degrading existing competencies for the EO tasks.

Our experimental results demonstrate the effectiveness of GeoVLM-R1 on multiple challenging EO tasks, as shown in Fig. 1. In particular, our method obtains a consistent improvement, highlighting the benefits of task-specific rewards, indicating robustness across EO tasks. The key contributions are summarized below:

- We develop GeoVLM-R1, a post-training RL framework tailored for reasoning capabilities in diverse EO tasks.

- We propose a novel dual-objective reward mechanism within GRPO, that introduces both format and correctness compliances, enhancing stable RL learning while producing accurate, structured, and interpretable reasoning paths.

- Experimental results on 28 downstream benchmarks show that our method performs well compared to existing VLMs and achieves better performance.

## 2 RELATED WORK

**Remote Sensing VLMs:** Recent advances in aligning visual and language data for remote sensing (RS) have led to the emergence of powerful Earth Observation (EO) vision–language models. RS-GPT (Hu et al., 2023) was the first to introduce an EO image–text paired dataset, enabling tasks such as image captioning and visual question answering (VQA). RemoteCLIP (Liu et al., 2024b) demonstrated strong zero-shot performance on classification and image–text retrieval. Models such as GeoChat (Kuckreja et al., 2024), SkyEyeGPT (Zhan et al., 2024), LHRS-Bot (Muhtar et al., 2024), and SkysenseGPT (Luo et al., 2024) extended these capabilities to region-level visual grounding through instruction-tuned, region-centric datasets and enhancing language understanding with LLMs. GeoPixel (Shabbir et al., 2025) further pushes the boundary to enable pixel-level grounding for the EO imagery. Beyond optical data, multimodal systems like EarthGPT (Zhang et al., 2024), EarthDial (Soni et al., 2025), and EarthMind (Shu et al., 2025) incorporated heterogeneous EO modalities for more comprehensive understanding. Despite these advances, current EO VLMs remain heavily reliant on supervised fine-tuning (SFT) and contrastive learning objectives (Khosla et al., 2020; Mall et al., 2023), which limits their robustness and restricts their reasoning capability.

**VLM Post-training:** Explicit post-training alignment techniques have been used to enhance general-purpose multimodal capabilities of VLMs, including prompt tuning (Liu et al., 2023; Zhu et al., 2023; Sheng et al., 2025) and reinforcement learning (RL) strategies (Huang et al., 2025; Shen et al., 2025; Guo et al., 2025). Among these, DPO (Rafailov et al., 2023) and PPO (Schulman et al., 2017) are widely adopted (Achiam et al., 2023; Chen et al., 2025; Tan et al., 2025; Deng et al., 2025b), where reward design plays a central role in guiding models toward producing coherent and structured outputs. However, traditional RL methods often suffer from high variance and unstable policy updates, particularly in complex structured reasoning tasks. To mitigate these challenges, group relative policy optimization (GRPO) (Shao et al., 2024), introduced in DeepSeek-R1 (Guo et al., 2025), leverages intra-group reward differences to stabilize training and improve structured reasoning (Peng et al., 2025; Tan et al., 2025; Deng et al., 2025a; Shen et al., 2025). However, the current reasoning models mainly focus on mathematical, coding, and general computer vision tasks, overlooking the potential of RL strategies in remote sensing tasks. An exception is UAV-VL-R1 (Guan et al., 2025), which applies RL to unmanned aerial vehicle imagery but is restricted to visual question answering (VQA). In contrast, the EO data encompasses a far broader spectrum of complex tasks in multi-sensory inputs (e.g., detection, captioning, grounding, change detection, and temporal analysis) that require more sophisticated post-training strategies capable of producing effective and interpretable reasoning paths.

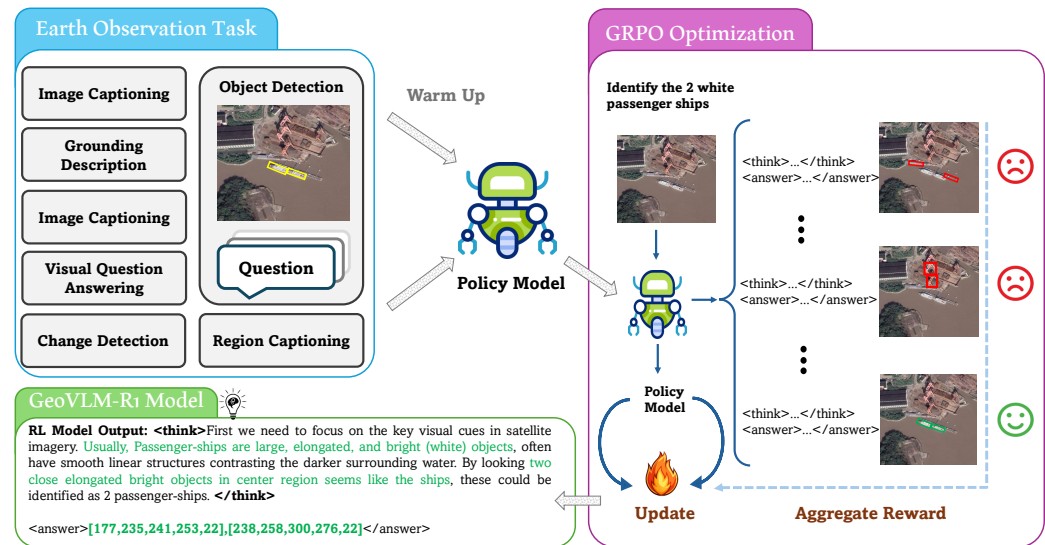

Figure 2: Illustration of the overall proposed training paradigm for GeoVLM-R1. The model is first initialized via supervised fine-tuning using diverse earth observation tasks. It is then successively optimized using GRPO-based reinforcement learning (RL) for each task. The GeoVLM-R1 processes queries and outputs a structured format that comprises an interpretable reasoning trace (<think>...</think>) and a final prediction (<answer>...</answer>).

## 3 METHOD

We propose GeoVLM-R1, a RL framework designed to enhance structured reasoning for complex EO tasks. Our method adopts a two-stage training paradigm (Fig. 2), combining supervised fine-tuning (SFT) with R1-style post-training based on GRPO (Shao et al., 2024). In the first stage, SFT equips the model with core EO knowledge and baseline reasoning ability by training across diverse tasks such as referred object detection, grounding, region captioning, classification, and temporal change detection. However, SFT alone yields shallow reasoning, often failing under complex multi-step EO queries. To address this limitation, we introduce a task-aware RL stage, where GRPO stabilizes optimization by exploiting relative advantages among candidate responses, while a dual-objective reward mechanism that enforces both semantic accuracy and structured interpretability that guides the model toward generating explicit reasoning traces before final predictions. This joint design allows GeoVLM-R1 to produce robust and interpretable reasoning paths that generalize effectively across diverse EO scenarios. We explain these training stages below.

### 3.1 SFT-BASED REASONING ACTIVATION

Given an EO multimodal sample $\mathcal{Q}_i = \{i, q_i\}$ consisting of a satellite image $i$ and corresponding text prompt $q_i$, the SFT training objective is to maximize the conditional likelihood of generating the target sequence $y_i$, which contains both reasoning steps and the final answer:

$$\mathcal{L}_{\text{SFT}}(\pi_\theta) = -\mathbb{E}_{(i,q_i,y_i)\sim\mathcal{D}} \left[ \sum_{t=1}^{T} \log \pi_\theta(y_{i,t} \mid i, q_i, y_{i,<t}) \right], \quad (1)$$

where $\mathcal{D}$ represents the training dataset, $\pi_\theta$ denotes the model with parameters $\theta$, and $y_{i,<t}$ represents the sequence of tokens generated before position $t$ for sample $i$. The resulting fine-tuned model $\pi_{\text{sft}}$ serves as a foundation for the subsequent reinforcement learning stage, ensuring the model has acquired fundamental EO domain knowledge and reasoning capabilities.

### 3.2 RL-BASED REASONING ENHANCEMENT

After SFT, we focus on enhancing the model's structured reasoning capabilities by leveraging analogous task-specific reward mechanisms through reinforcement learning. In contrast to traditional

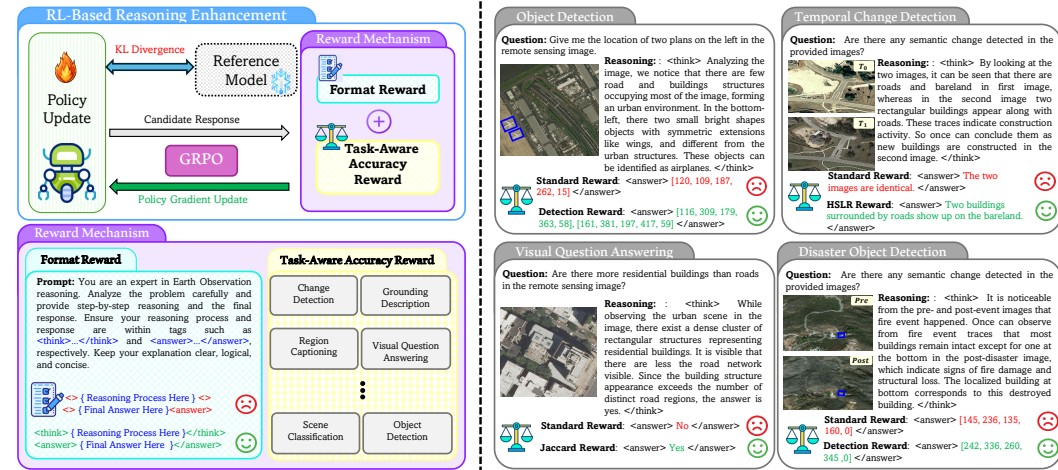

Figure 3: Overall pipeline of GeoVLM-R1 policy update mechanism (left). During fine-tuning, the GRPO module generates multiple candidate responses. These responses are evaluated, and each is assigned a distinct reward equipped with our reward mechanism. In particular, our reward mechanism comprises (i) a format reward to enforce structural compliance and (ii) a task-aware accuracy reward to ensure accuracy compliance. We present a few examples showcasing GeoVLM-R1 using a unique task-aware accuracy reward function, resulting in better performance (right).

PPO, which requires an additional critic model to estimate policy performance and incurs high computational cost, we employ GRPO that mitigates the need for a separate critic by directly utilizing relative rewards among candidate responses, making it particularly effective for structure-aware and constraint-driven visual reasoning tasks.

Given a multimodal sample $\mathcal{Q}_i$, GRPO generates a group of $K$ candidate responses $S_{\mathcal{Q}_i} = \{s_1, s_2, \ldots, s_K\}$ from the old policy model $\pi_{\theta_{\text{old}}}$. The current policy model $\pi_\theta$ is then optimized using the following objective:

$$\mathcal{J}_{\text{GRPO}}(\theta) = \mathbb{E}_{\{s_i\}_{i=1}^K \sim \pi_{\theta_{\text{old}}}(\mathcal{Q}_i)} \left[ \frac{1}{K} \sum_{i=1}^K \min \left[ \rho_i A_i, \text{clip}(\rho_i, 1 - \epsilon, 1 + \epsilon) A_i \right] - \beta D_{\text{KL}}[\pi_\theta \| \pi_{\text{ref}}] \right] \quad (2)$$

$$\rho_i = \frac{\pi_\theta(s_i | \mathcal{Q}_i)}{\pi_{\theta_{\text{old}}}(s_i | \mathcal{Q}_i)}, \quad D_{\text{KL}}[\pi_\theta \| \pi_{\text{ref}}] = \mathbb{E}_{s \sim \pi_\theta} \left[ \log \frac{\pi_\theta(s | \mathcal{Q}_i)}{\pi_{\text{ref}}(s | \mathcal{Q}_i)} \right] \quad (3)$$

where the policy ratio $\rho_i$ controls the update step size, $\epsilon$ denotes the clipping threshold, and $\beta$ controls the strength of the KL penalty (Schulman et al., 2015; 2017) that prevents $\pi_\theta$ from deviating excessively from the reference model $\pi_{\text{ref}}$. For each candidate response $s_i$, an analogous task-specific reward function $r_i = R(\mathcal{Q}_i, s_i)$ quantifies the quality of the candidate response in the context of the given sample. GRPO computes the relative advantage $A_i$ for candidate response $s_i$ compared to other responses as:

$$A_i = \frac{r_i - \bar{r}}{\sigma_r} \quad (4)$$

where $\bar{r} = \frac{1}{K} \sum_{j=1}^K r_j$ is the mean reward and $\sigma_r = \sqrt{\frac{1}{K} \sum_{j=1}^K (r_j - \bar{r})^2}$ is the standard deviation across all candidate responses. This normalization process reduces reward variance across samples, thereby stabilizing training and enhancing the robustness of policy gradient estimation.

### 3.3 TASK-AWARE REWARD DESIGN FOR VISUAL REASONING

Inspired by recent progress in applying RL to enhance reasoning capabilities (Shao et al., 2024; Shen et al., 2025), we adopt an RL-based post-training strategy to enhance the reasoning capabilities of the policy model. In contrast to mathematics and coding tasks where ground-truth is well-defined, the EO data samples pose unique challenges in reward design for various tasks. Therefore, as can be seen in Fig. 3, we have a sophisticated reward mechanism, enabling effective RL in EO reasoning contexts. To generate structurally coherent and semantically accurate reasoning outputs, we introduce format and task-aware accuracy rewards to better guide reasoning optimization.

Table 1: Summary of QA instruction pairs and reward functions used in GRPO optimization across diverse Earth Observation tasks.

| Dataset | Temporal | Task | # QA Pairs | Task-Aware Accuracy Rewards |
|---|---|---|---|---|
| BigEarthNet (Sumbul et al., 2019) | Single | Classification | 30,000 | Recall |
| RSCIS (Lu et al., 2017) | Single | Image Captioning | 43,670 | Levenshtein Similarity Ratio |
| RSVQA-LRBEN (Lobry et al., 2020) | Single | Visual Question Answering | 57,223 | Jaccard |
| GeoChat-Instruct (Kuckreja et al., 2024) | Single | Region Captioning | 69,269 | SBERT |
| GeoChat-Instruct (Kuckreja et al., 2024) | Single | Referred Object Detection | 73,000 | Detection |
| GeoChat-Instruct (Kuckreja et al., 2024) | Single | Grounding | 69,269 | Lexical-Metric-based Grounding Reward |
| xBD (Gupta et al., 2019) | Bi-Temporal | Referred Object Detection | 4,202 | Detection |
| xBD (Gupta et al., 2019) | Bi-Temporal | Object Detection | 2,283 | Detection |
| LEVIR-MCI (Liu et al., 2024a), DUBAI-CC[1], MUDS (Yang et al., 2024) | Bi-/Multi-Temporal | Change Detection Caption | 352,825 | Hybrid SBERT and Lexical-Metric |
| FMoW (Irvin et al., 2024) | Multi-Temporal | Classification | 83,412 | Accuracy |

**Format Reward:** The objective of format reward ($R_{\text{format}}$) is to make sure that the model's output adheres to a predefined structured format. It comprises (i) *think reward*, intending to think deeply before answering and constrain the model to have `<think>`$t$`</think>` tags, where $t$ is the language reasoning, and (ii) an *answer reward* to generate the final answer $a$ having `<answer>`$a$`</answer>` tags. If both reward tags are included in the response, the reward is 1; otherwise, it is 0.

**Task-aware Accuracy Reward:** The goal of this reward ($R_{\text{task\_acc}}$) is to quantify the semantic correctness of the content ($a$) within the `<answer></answer>`, matches with the ground-truth answer $g_i$. Hence, the total reward is defined as: $R(a) = R_{\text{format}} + R_{\text{task\_acc}}$, where $R_{\text{task\_acc}} \in [0, 1]$. Table 1 presents datasets, tasks, the number of question-answer pairs for each task, and the reward functions used for each task during RL process. Now, we present the details of task-aware accuracy reward functions.

**Recall Reward:** We employ recall as a reward function in RL fine-tuning of a vision-language model for the classification task. It is important to detect rare but critical instances, particularly in disaster assessment scenarios. To encourage the sensitivity to correct positive predictions for classification tasks, we define a recall reward as: $R_{\text{Recall}} = \frac{\text{TP}}{\text{TP}+\text{FN}}$, where TP is the number of true positives and FN is the number of false negatives.

**Sentence-BERT (SBERT) Reward:** The region-captioning task describes the complex visual content, demanding the model to output key semantic elements (category, color, relative size, relative location, position) even if phrased differently. To capture the semantic fidelity between the candidate response and ground truth strings, we employ a Sentence-BERT (SBERT)-based reward function (Reimers & Gurevych, 2019). We encode each string into a fixed-dimensional embedding such that semantically similar strings exhibit high cosine similarity. Let $\mathbf{e}_{s_i}$ and $\mathbf{e}_{g_i}$ represent the embeddings of the candidate response and ground truth string, respectively. The SBERT reward is defined as: $R_{\text{SBERT}} = \max\left(0, \cos(\mathbf{e}_{s_i}, \mathbf{e}_{g_i})\right) = \max\left(0, \frac{\mathbf{e}_{s_i} \cdot \mathbf{e}_{g_i}}{\|\mathbf{e}_{s_i}\| \|\mathbf{e}_{g_i}\|}\right)$, where $\cos(\cdot, \cdot)$ represents the cosine similarity function. Since cosine similarity ranges from $-1$ to 1, we apply a rectified linear transformation to ensure $R_{\text{SBERT}} \in [0, 1]$, which prevents negative rewards and maintains compatibility with our RL objectives.

**Detection Reward:** To evaluate the precise spatial accuracy for the object detection task, where the model outputs a rotated bounding box, we formulate the reward function based on the Intersection-over-Union (IoU) between the candidate response and the ground-truth rotated bounding box. We compute the final reward by computing a matching reward by pairing each ground truth bounding box with the best-overlapping predicted bounding box as: $R_{\text{Detection}} = \frac{1}{N} \sum_{n=1}^{N} \max_m \text{IoU}(s_i{}^m, g_i{}^n)$, where $N$ is the total number of ground truth. This reward encourages the RL model to generate bounding boxes that closely match the ground truth bounding boxes.

**Lexical-Metric-based Grounding Reward (LMGR):** The grounding description task comprises both object detection and text description, which requires a hybrid reward function to force the RL model to learn both object detection as well as textual description, aligning semantically. Using detection reward alone ignores the quality of text description and vice versa, leading to performance degradation. For spatial accuracy, we use $R_{\text{Detection}}$. For semantic correction, to evaluate the lexical accuracy and informativeness of the string, we employ an average of Rouge-1 (R1), Rouge-L (RL), and Meteor (MT) metrics. The reward is defined as: $R_{\text{LM}} = \frac{\alpha\,R_1 + \beta\,R_L + \gamma\,R_{\text{MT}}}{3}$ where $\alpha$, $\beta$, and $\gamma$ are set to 1. Finally, we combine $R_{\text{Detection}}$ and $R_{\text{LM}}$ to encode the spatial grounding and lexical fidelity, and it can be expressed as $R_{\text{LMGR}} = \frac{R_{\text{LM}} + R_{\text{Detection}}}{2}$.

**Levenshtein Similarity Ratio (LR) Reward:** The image caption task requires the model to provide a sequence-level similarity, which is structured and worded to human references. Therefore,

| Model | AID (ZS) | UCMerced (ZS) | WHU-19 (ZS) | BigEarthNet | xBD Set 1 (Temporal) | FMoW (Temporal) |
|---|---|---|---|---|---|---|
| GPT-4o | 74.73 | 88.76 | 91.14 | 49.00 | 67.95 | 21.43 |
| InternVL-8B Chen et al. (2024) | 60.40 | 58.23 | 79.30 | 19.73 | 51.44 | 21.04 |
| Qwen2.5-VL-3B Bai et al. (2025) | 58.27 | 60.86 | 78.21 | 24.75 | 51.44 | 34.36 |
| GeoChat Kuckreja et al. (2024) | 72.03 | 84.43 | 80.09 | 20.35 | 53.32 | 59.2 |
| EarthDial Soni et al. (2025) | 88.76 | 92.42 | 96.21 | 73.03 | 96.37 | 70.03 |
| **GeoVLM-R1** | 88.46 | 97.81 | 97.91 | 80.91 | 98.93 | 76.93 |

Table 2: GeoVLM-R1 illustrates a consistent improvement among zero-shot (ZS), multi-label BigEarthNet, and temporal classification datasets compared to other existing VLMs.

we employ Levenshtein similarity ratio (Po, 2020), where we quantify the similarity between the candidate response $s_i$ and ground truth $g_i$, going beyond binary correctness and supporting partial credit for near matches. The reward function is defined as: $R_{\text{LR}} = \frac{|s_i|+|g_i|-D(s_i,g_i)}{|s_i|+|g_i|}$, where $|s_i|$ and $|g_i|$ denote the length of strings and $D(s_i, g_i)$ is the Levenshtein distance. The $R_{\text{LR}} \in [0, 1]$ with a value of 0 indicates totally dissimilar image captions, and a value of 1 means that two captions are identical.

**Jaccard Similarity Reward:** The visual question answering (VQA) task outputs short phrases; therefore, giving partial credit for answers is important, rather than requiring exact matches. We employ a Jaccard similarity reward function, which measures the ratio of the intersection to the union between candidate response and ground truth tokens. It is defined as: $R_{\text{Jaccard}}(s_i, g_i) = \frac{|s_i \cap g_i|}{|s_i \cup g_i|}$.

**Hybrid SBERT and Lexical-Metric Reward (HSLR):** Change detection task involves the textual description between the pre-change and post-change events in the scene. The textual description indicates the semantic changes that occurred, such as the construction or demolition of roads, buildings, or any man-made infrastructure. The RL goal is to align visual observations with their corresponding language expressions. To leverage both semantic fidelity and lexical accuracy, we define a hybrid reward combining $R_{\text{SBERT}}$ and $R_{\text{LM}}$. This hybrid reward is defined as: $R_{\text{HSLR}} = \frac{R_{\text{SBERT}}+R_{\text{LM}}}{2}$.

## 4 EXPERIMENTS

### 4.1 IMPLEMENTATION DETAILS

We select Qwen2.5VL-3B-Instruct (Bai et al., 2025) as our base model due to its promising performance on visual-language understanding. We adopt the EarthDial-Instruct (Soni et al., 2025) and resized the images to $448 \times 448$ before passing to the model and normalized the rotated bounding boxes between 0-448 to ensure consistency across the multi-resolution images.

For SFT, we fine-tune the model using 8 A100 GPUs for 2 epochs, setting the batch size to 2 per device, the learning rate to $1e-5$, weight decay to 0.1, and a warmup ratio of 0.03 under a cosine learning rate scheduler. For GRPO, we use 4 A100 GPUS and fine-tune for 2 epochs with batch size = 1, gradient accumulation = 2, bfloat16 precision, temperature to 0.9, KL divergence ratio (i.e., $\beta$) to 0.04, and learning rate of $1e-6$. Following (Soni et al., 2025), we discuss the results of our method in a diverse set of applications for RS optical imagery, such as scene classification, region captioning, refer object detection, grounding descriptions, VQA, image captioning, and temporal change detection captioning.

### 4.2 STATE OF THE ART COMPARISONS

**Scene Classification:** Table 2 compares our method with existing VLMs over diverse scene classification datasets. We notice that our method shows an improvement over the zero-shot evaluation. In addition, our method achieves 7.88% improvement compared to recent EarthDial over the large-scale multi-label BigEarthNet dataset. Moreover, our method shows promising results over temporal datasets. For instance, our method gains an absolute advantage of 2.56% and 6.9% over xBD testset-1 and FMoW datasets, respectively.

**Referred Object Detection, Region Captioning, and Grounding Descriptions:** In Table 3, we compare GeoVLM-R1 over three tasks (including referred object detection, region captioning, and grounding description). For the referred object detection task, our method consistently shows better results by a large margin. For example, for multiple object detection, we obtain 21.63% gain

| | Referred Object Detection Task | | | | | | | | | | Region-Captioning Task | | | | | | Grounding Task | | | | |
| | GeoChat-Instruct | | | | | NWPU VHR-10 (Zero-Shot) | | | | | GeoChat-Instruct | | | NWPU VHR-10 (Zero-Shot) | | | NWPU VHR-10 (Zero-Shot) | | | | |
| | Small | Med. | Large | Single | Mult. | Small | Med. | Large | Single | Mult. | Rouge1 | Rouge-L | Meteor | Rouge1 | Rouge-L | Meteor | @0.5 | @0.25 | Rouge1 | Rouge-L | Meteor |
|---|---|---|---|---|---|---|---|---|---|---|---|---|---|---|---|---|---|---|---|---|---|
| GPT-4o | - | - | - | - | - | - | - | - | - | - | 9.41 | 7.6 | 8.02 | 17.68 | 11.81 | 9.63 | 0.7 | 6.1 | 14.72 | 10.82 | 9.41 |
| InternVL2-4B | 6.3 | 24.37 | 37.38 | 24.96 | 11.72 | 7.1 | 12.68 | 25.48 | 22.96 | 8.1 | - | - | - | - | - | - | 10.6 | 29.87 | 30.67 | 29.09 | 21.92 |
| InternVL2-8B | 7.20 | 23.76 | 31.99 | 25.77 | 9.30 | 4.26 | 11.85 | 20.72 | 21.66 | 5.86 | 10.58 | 9.06 | 8.5 | 11.88 | 9.63 | 7.7 | - | - | - | - | - |
| GeoChat | 2.9 | 13.6 | 21.7 | 16 | 4.3 | 3.2 | 14.7 | 13.23 | 1.9 | | 72.77 | 72.74 | 61.9 | 62.02 | 62.02 | 53.31 | 2.2 | 15.27 | 21.46 | 20.74 | 21.38 |
| EarthDial | 11.43 | 31.76 | 39.07 | 34.29 | 13.41 | 11.66 | 14.21 | 23.12 | 25.37 | 8.9 | 73.38 | 73.34 | 62.72 | 72.14 | 72.14 | 60.01 | 17.07 | 41.00 | 27.05 | 26.35 | 23.12 |
| **GeoVLM-R1** | 36.02 | 54.72 | 55.03 | 57.1 | 35.04 | 34.44 | 48.76 | 64.91 | 55.97 | 41.45 | 75.92 | 75.9 | 66.43 | 72.10 | 72.10 | 55.49 | 38.74 | 61.45 | 31.34 | 30.08 | 26.10 |

Table 3: GeoVLM-R1 illustrating a consistent performance gain, indicating better capabilities to locate objects, across referred object detection, region-captioning, and grounding description tasks.

| Model | CD Dubai-CC | | | CD LEVIR-MCI | | | CD MUDS | | | CD SYSU (ZS) | | | IC NWPU-Captions | | | IC RSCID-Captions | | | IC RSITMD-Captions (ZS) | | |
| | Rouge1 | Rouge-L | Meteor | Rouge1 | Rouge-L | Meteor | Rouge1 | Rouge-L | Meteor | Rouge1 | Rouge-L | Meteor | Rouge1 | Rouge-L | Meteor | Rouge1 | Rouge-L | Meteor | Rouge1 | Rouge-L | Meteor |
|---|---|---|---|---|---|---|---|---|---|---|---|---|---|---|---|---|---|---|---|---|---|
| GPT-4o | 8.81 | 7.45 | 18.68 | 10.33 | 8.4 | 22.05 | 14.18 | 11.02 | 20.92 | 16.48 | 12.32 | 17.49 | 19.43 | 14.86 | 28.16 | 20.53 | 15.59 | 26.03 | 18.31 | 14.22 | 24.83 |
| InternVL2-4B Chen et al. (2024) | 7.31 | 6.38 | 21.12 | 8.88 | 7.43 | 22.14 | 10.25 | 7.90 | 17.73 | 13.27 | 9.98 | 14.36 | 0 | 0 | 0 | 0 | 0 | 0 | 0 | 0 | 0 |
| InternVL2-8B Chen et al. (2024) | - | - | - | - | - | - | - | - | - | - | - | - | 20.69 | 15.64 | 30.18 | 21.59 | 16.13 | 28.17 | 18.91 | 14.65 | 26.02 |
| Qwen2.5-VL-3B Bai et al. (2025) | 14.41 | 13.62 | 27.59 | 12.27 | 10.11 | 26.11 | 12.13 | 9.30 | 18.22 | 13.61 | 10.34 | 16.06 | 18.82 | 14.72 | 26.79 | 21.37 | 16.42 | 26.53 | 18.79 | 15.02 | 25.05 |
| GeoChat Kuckreja et al. (2024) | 14.21 | 14.39 | 28.91 | 17.15 | 35.42 | 12.35 | 12.28 | 12.23 | 15.98 | 13.45 | 12.02 | 13.96 | 14.86 | 12.54 | 15.21 | 13.48 | 11.59 | 12.39 | 13.41 | 11.50 | 12.33 |
| EarthDial | 31.94 | 30.66 | 55.83 | 33.78 | 30.47 | 74.8 | 28.16 | 24.03 | 33.56 | 18.03 | 17.42 | 14.98 | 45.84 | 39.96 | 80.61 | 33.77 | 27.61 | 56.18 | 26.74 | 21.72 | 34.06 |
| **GeoVLM-R1** | 36.60 | 34.15 | 61.22 | 37.85 | 34.02 | 73.56 | 34.07 | 27.65 | 45.94 | 19.64 | 18.46 | 15.45 | 46.94 | 40.96 | 82.00 | 34.64 | 28.63 | 56.54 | 30.62 | 25.39 | 39.07 |

Table 4: Comparison of GeoVLM-R1 over change detection (CD) and image captioning (IC) datasets. Results indicate better capabilities of our method to generate captions compared to existing VLMs for both temporal CD and image-captioning datasets. ZS means zero-shot evaluation.

| Model | Image Captioning | | | Region Classification | | Image Classification | | | Object Detection | | Referred Object Detection | |
| | Rouge1 | Rouge-L | Meteor | Test Set-1 | Test Set-2 | Test Set-1 | Test Set-2 | Test Set-3 | mAP@0.5 | mAP@0.25 | mAP@0.5 | mAP@0.25 |
|---|---|---|---|---|---|---|---|---|---|---|---|---|
| GPT-4o | 14.21 | 10.35 | 19.52 | 51.68 | 71.62 | 67.95 | 75.45 | 70.41 | 0.2 | 2.15 | 0 | 0 |
| InternVL2-8B | 13.89 | 10.37 | 14.92 | 14.39 | 58.33 | 51.44 | 61.52 | 51.12 | 0.6 | 1.07 | 0 | 0.7 |
| Qwen2.5-VL-3B | 11.98 | 8.12 | 19.94 | 71.19 | 59.69 | 51.44 | 56.16 | 41.26 | - | - | - | - |
| GeoChat | 14.18 | 10.67 | 12.20 | 25.30 | 57.65 | 53.32 | 52.19 | 49.51 | 1.15 | 7.2 | 0.2 | 3.09 |
| EarthDial | 87.26 | 87.26 | 88.53 | 53.70 | 83.09 | 96.37 | 82.85 | 54.01 | 7.6 | 21.11 | 5.1 | 13.09 |
| **GeoVLM-R1** | 92.26 | 92.26 | 93.37 | 81.36 | 83.55 | 98.93 | 86.39 | 68.60 | 38.15 | 48.13 | 24.52 | 34.52 |

Table 5: We compare GeoVLM-R1 for various tasks on the xBD dataset (temporal). Our method exhibits substantial progress across the tasks. In particular, our approach shows a notable performance gain over object detection and referred object detection tasks, compared to other VLMs.

compared to EarthDial. Our method demonstrates a consistent improvement across the tasks over GeoChat-Instruct and NWPU VHR-10 datasets. In the case of the region captioning task, our method obtains better performance compared to other methods over GeoChat-Instruct and comparable performance over the NWPU-VHR-10 dataset. Furthermore, our method presents a promising performance in the grounding description task, particularly in object detection, where other VLMs struggle to localize the objects. In short, we notice that our approach shows favorable performance in these tasks, demonstrating its merits.

**Image Captioning, and Change Detection Captioning:** Our GeoVLM-R1 shows consistent performance gain across the image captioning datasets as shown in Table 4. Similarly, it consistently performs favorably against the existing generic and specialized VLMs over change detection captioning datasets.

**Temporal Disaster Assessment :** We also validate the performance of our GeoVLM-R1 over the temporal building damage assessment xBD dataset (Gupta et al., 2019) in the Table 5. This dataset covers eight diverse tasks, such as tem-

| Model | Presence | Comp | R/U | Avg. | Model | Presence | Comp | Avg. |
|---|---|---|---|---|---|---|---|---|
| MiniGPTv2 | 55.16 | 55.22 | 39.00 | 54.96 | MiniGPTv2 | 40.79 | 50.91 | 46.46 |
| Qwen2-VL | 38.57 | 67.59 | 61.00 | 55.35 | Qwen2-VL | 66.44 | 60.41 | 63.06 |
| InternVL2-8B | 58.54 | 72.28 | 71.00 | 66.51 | InternVL2-8B | 67.35 | 76.91 | 72.70 |
| Qwen2.5-VL-3B | 59.59 | 75.04 | 63.00 | 68.40 | Qwen2.5-VL-3B | 59.89 | 72.26 | 66.81 |
| GeoChat | 91.09 | 90.33 | 94.00 | 90.70 | GeoChat | 58.45 | 83.19 | 72.30 |
| LHRS-Bot | 88.51 | 90.00 | 89.07 | 89.19 | EarthGPT | 62.77 | 79.53 | 72.06 |
| TeoChat | 91.70 | 92.70 | 94.00 | 92.29 | TeoChat | 67.50 | 81.10 | 75.04 |
| EarthDial | 92.58 | 92.75 | 94.00 | 92.70 | EarthDial | 58.89 | 83.11 | 72.45 |
| **GeoVLM-R1** | 91.81 | 93.20 | 96 | 92.66 | **GeoVLM-R1** | 66.38 | 82.26 | 75.27 |

Table 6: Our method performs better compared to existing VLMs for Comp and R/U categories over RSVQA-LRBEN (left) and obtains a better average score for RSVQA-HRBEN (right). Comp: Comparison, R/U: Rural/Urban.

poral image captioning, region classification, image classification, object detection, and referred object detection. Our method is compared with recent EarthDial and other generic and specialized VLMs. Overall, our method demonstrates better performance across tasks. In addition, our method achieves significant performance gain over object detection and referred object detection tasks, where recent EarthDial as well as existing VLMs struggle alot. For example, in the case of object detection, our approach obtains an absolute gain of 30.55% using the mAP@0.5 metric, which demonstrates the effectiveness of our method.

**Visual Question Answering (VQA):** We demonstrate the performance of our method on the VQA task in Table 6. Following (Soni et al., 2025), we compare our method over RSVQA-LRBEN and RSVAQ-HRBEN. Our method demonstrates advantages for comparison and the rural/urban category over RSVQA-LRBEN. Moreover, in the RSVQA-HRBEN dataset, our method achieves a better weighted average score of 75.27% using zero-shot evaluation.

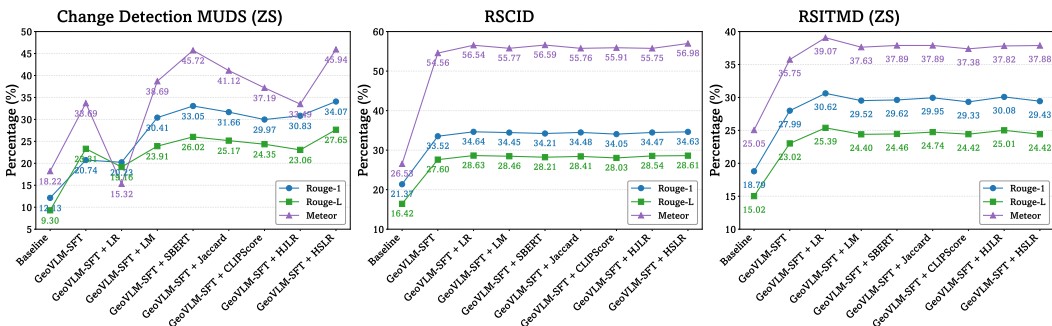

Figure 4: Ablation over change detection MUDS dataset shows that GeoVLM-R1 with HSLR performs better. Whereas for image captioning task, GeoVLM-R1 with LR reward performs favorably.

## 4.3 ABLATION STUDY

To validate the effectiveness of our GeoVLM-R1 using task-aware accuracy reward-based during GRPO optimization, we perform extensive ablation experiments across various tasks as discussed below. To do so, we first fine-tune our base model (Qwen2.-VL-3B) using EarthDial-Instruct to obtain GeoVLM-SFT and then apply the proposed R1-style optimization across tasks. To validate our RL-based approach, we employ different reward functions (e.g., Levenshtein Ratio (LR), Lexical-Metric (LM), Jaccard, Detection Reward, Recall, Sentence-BERT (SBERT), Hybrid SBERT and Lexical-Metric Reward (HSLR), Hybrid Jaccard and Lexical-Metric Reward (HJLR), SBERT-based Grounding Reward (SGR), and Lexical-Metric-based Grounding Reward (LMGR)).

**Ablation on Classification Tasks:** During RL process, we introduce a range of reward functions, such as Jaccard, Levenshtein Ratio (LR), CLIPScore, SBERT, accuracy, and recall reward functions for the classification task. As in Fig. 5, GeoVLM-SFT achieves 73.03. Using the recall reward function, our GeoVLM-R1 achieves higher results than all other methods with an 80.91% score.

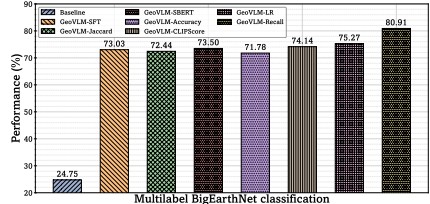

**Image Captioning and Change Detection:** Fig. 4 indicates that GeoVLM-R1 utilizing LR reward function performs better compared to other reward functions using zero-shot evaluation over RSITMD-Captions dataset. Similarly, in case of the change detection captioning task, the proposed HSLR reward shows better score across metrics (e.g., Rouge-1, Rouge-L, and Meteor) over MUDS zero-shot evaluation.

Figure 5: Ablation using various reward functions for the classification task. Our method with the recall reward is more effective than other models.

**Ablation on Referred Object Detection, Region-Captioning, and Grounding Description Tasks:** The baseline does not provide the rotated bounding boxes (RBB). For fair comparison, we fine-tune our baseline using RBB and then apply accuracy-aware reward policy optimization. For the referred object detection task, we apply a detection reward, where the output responses and ground truth are first converted to polygons and compared based on their IoU score. In addition, we also apply the detection reward using horizontal bounding boxes (HBB), where the boxes are first converted to horizontally aligned boxes and set the angle zero. We observe that small-angle prediction errors can reduce IoU during

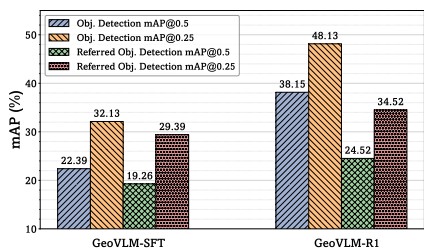

Figure 6: Ablation on xBD shows GeoVLM-R1 improves building localization.

RL. Therefore, during GRPO optimization, HBB predictions with higher intragroup advantage guide the policy toward more stable and reward-maximizing outputs. For the region captioning task, we notice that SBERT reward performs better. In case of the grounding description task, it is crucial to correctly locate the objects and provide their description. We notice that our RL approach using LMGR shows significant improvement using zero-shot evaluation, as shown in Fig. 7.

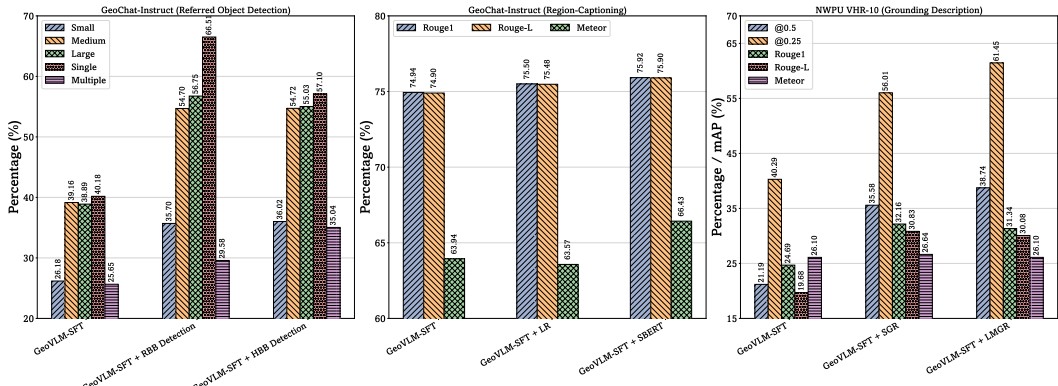

Figure 7: Ablation over referred object detection shows that horizontally aligned boxes during RL result in better object detection. GeoVLM-R1 with SBERT and LMGR reward functions performs better for region-captioning and grounding description tasks, respectively.

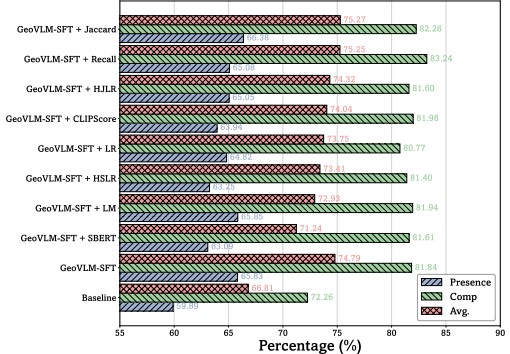

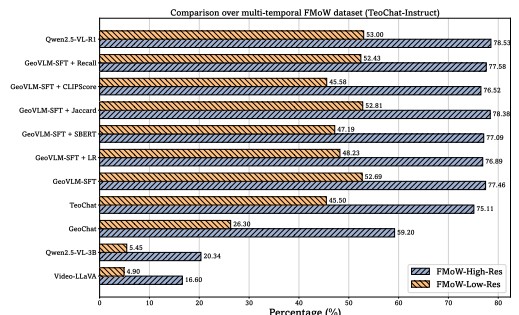

Figure 8: Ablation over RSVQA-HRBEN dataset. The zero-shot evaluation shows that our method (GeoVLM-R1) with Jaccard reward achieves a better score over the presence category as well as the favorable average score.

Figure 9: Comparison over multi-temporal FMoW dataset, where the model is fine-tuned and tested over TeoChat-Instruct. GeoVLM-R1 with the accuracy reward performs favorably against other VLMs.

**VQA and xBD Object Detection:** In Fig. 6, we show that our method GeoVLM-R1 performs better and obtains a significant gain compared to GeoVLM-SFT, indicating the effectiveness of GRPO optimization using detection reward on the xBD dataset. Moreover, for the VQA task, the Jaccard similarity reward function performs better over the presence category and obtains a better average score compared to other methods in Fig. 8 over RSVQA-HRBEN using zero-shot evaluation, which reflects the merits of GeoVLM-R1.

**Experiments on FMoW:** We also demonstrate our method's performance on the temporal FMoW dataset using TeoChat-Instruct (Irvin et al., 2024) as shown in Fig. 9. Experimental results reveal that our GeoVLM-R1 using an accuracy-based reward function obtains a favorable score over both FMoW-High-Res and FMoW-Low-Res datasets.

## 5 CONCLUSION

In this work, we present GeoVLM-R1, an effective post-training framework tailored for task-oriented structured reasoning in remote sensing imagery. To mitigate the poor reasoning capabilities of domain-specific VLMs, we propose supervised finetuning and subsequently task-oriented-based GRPO reinforcement learning, where a task-aware accuracy reward function is combined with format reward to minimize the policy variance and improve the stable, structured, and semantically consistent reasoning path. Extensive experiments demonstrate that our reinforcement learning approach is effective and obtains state-of-the-art performance across EO tasks.

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
