# OpenReview forum: "GeoVLM-R1: Reinforcement Fine-Tuning for Improved Remote Sensing Reasoning"
_ICLR.cc/2026/Conference — ICLR 2026 Conference Withdrawn Submission_

### Official Review · Reviewer_q2vs · 2025-10-31

**Soundness:** 3
**Presentation:** 3
**Contribution:** 3
**Rating:** 6
**Confidence:** 2

**Summary:**

This paper proposes GeoVLM-R1, a geo-observation multimodal vision-language model that combines SFT with R1-style reinforcement learning. The authors first use SFT to quickly align the model with the target distribution, and then apply a GRPO-based reinforcement optimization strategy together with a task-aware dual reward mechanism, achieving consistent performance improvements across multiple remote sensing tasks such as classification, detection, description, and temporal analysis.

**Strengths:**

1.	The paper introduces a GRPO-based reinforcement optimization framework, which replaces traditional PPO/DPO methods. By leveraging group-relative advantages, it reduces training variance and stabilizes the reasoning process. Moreover, the authors design task-specific reward functions, e.g., Recall reward for classification, IoU reward for detection, and hybrid semantic rewards for description tasks.
2.	It incorporates a dual-objective reward that not only ensures semantic correctness but also enforces the generation of explicit reasoning chains and structured answers, improving interpretability and transparency.
3.	GeoVLM-R1 serves as a general post-training framework that can be seamlessly extended to classification, detection, description, change detection, and disaster assessment tasks. The training process is stable and scalable, and the model achieves superior performance on the multi-task EarthDial-Bench benchmark compared with existing VLMs.

**Weaknesses:**

1.	The task rewards rely heavily on textual/semantic similarity and format-compliance signals, which may not fully constrain the strict geometric correctness required in geospatial contexts. The authors also convert rotated bounding boxes into horizontal ones to alleviate small-angle IoU penalties-this stabilizes optimization but may weaken sensitivity to real geometric accuracy, potentially leading to reward overfitting rather than true task generalization.
2.	GeoVLM normalizes all imagery to 448x448 and scales bounding boxes accordingly. While this improves efficiency, it may lose critical information for small or fine-grained targets common in remote sensing. The work does not quantify how different resolutions affect performance, particularly for detection and change localization tasks.

**Questions:**

see above weakness

---

### Official Review · Reviewer_EhnT · 2025-10-31

**Soundness:** 2
**Presentation:** 2
**Contribution:** 2
**Rating:** 4
**Confidence:** 3

**Summary:**

This paper presents GeoVLM-R1, a reinforcement learning fine-tuning framework for EO tasks, aiming to enhance the structured reasoning ability of VLMs in complex EO scenarios.
The key contribution lies in the design of a dual-objective reward mechanism, which forces the model to engage in reasoning through format compliance rewards and improves the model's adaptability to different tasks through task-aware rewards. And it has demonstrated effectiveness and robustness in multiple EO tasks.

**Strengths:**

This work is the first to systematically introduce task-aware RL fine-tuning into the multi-task remote sensing scenario, filling the gap of RL in the EO field. The experimental design is reasonable, covering a wide range of remote sensing tasks and datasets. The method description is clear, and the reward function design is specific and reproducible. It addresses the current weak reasoning ability of EO-VLMs and promotes the development of remote sensing multimodal understanding towards structured and interpretable directions.

**Weaknesses:**

1.	Although the reward function is task-aware, it must be manually defined for each task type. The authors do not provide a general strategy or guidelines for selecting an appropriate reward function when encountering new tasks not evaluated in the paper—such as quantitative analyses involving spectral data for vegetation monitoring, temperature retrieval, or drought assessment.
2.	The authors only compare different task-aware reward functions but fail to investigate whether deep reasoning is truly necessary for each task. An ablation study examining the impact of explicitly enforcing or disabling reasoning capabilities would strengthen the analysis.
3.	Directly summing the format reward (0 or 1) with the task reward (ranging from 0 to 1) may bias the model toward over-optimizing formatting correctness at the expense of semantic accuracy. The authors should carefully address this issue and explore more robust strategies for combining these two reward components.
4.	While the authors demonstrate the effectiveness of their reinforcement learning framework, they lack a systematic discussion of scenarios where the model underperforms—such as cases involving densely packed small objects, extreme lighting conditions, or cross-sensor data generalization. This limits the understanding of the model’s robustness boundaries.
5.	In the ablation studies, the authors compare model performance under different reward functions for specific tasks, but their analysis remains limited to presenting quantitative results in tables and figures. They do not delve into why a particular reward function works better for a given task or provide insights into the underlying reasons for performance differences among reward functions.

**Questions:**

1.	How should K (the number of candidate responses) be chosen in GRPO? Does K need to be adjusted for different tasks?
2.	In Figure 8, the model GeoVLM-SFT, which is fine-tuned only with SFT and without any RL strategy, achieves average performance that even surpasses most models trained with RL using various reward functions. How should this phenomenon be interpreted?
3.	Figure 4 presents ablation studies for the Image Captioning and Change Detection tasks using line plots, which may not be sufficiently intuitive.
4.	Could the format reward potentially suppress the model’s expressiveness? Does enforcing the <think>/<answer> structure introduce redundancy or limit flexibility in certain tasks?

---

### Official Review · Reviewer_GvFx · 2025-10-31

**Soundness:** 2
**Presentation:** 2
**Contribution:** 2
**Rating:** 2
**Confidence:** 3

**Summary:**

The paper proposes GeoVLM-R1, a post-training recipe for remote-sensing VLMs that couples SFT and GRPO with a dual objective: a format reward enforcing <think>…</think> / <answer>…</answer> structure and task-aware rewards aligned to downstream supervision (e.g., IoU/mAP for (referred) detection and grounding, a lexical–semantic mix for region/change captions, Levenshtein for image captioning, Jaccard for short-answer VQA, and recall/accuracy for scene and multi-temporal classification). The framework aims to make reasoning verifiable by optimizing directly on these task metrics, and is evaluated across classification, referred detection/grounding, captioning (image/region/change), VQA, and temporal analysis.
Across 28 benchmarks, the model reports consistent improvements—most pronounced on spatial localization and multi-temporal datasets (e.g., notable gains on xBD, FMoW, and BigEarthNet), with large margins for multi-object referred detection and disaster-scene detection.

**Strengths:**

1. Broad task coverage with strong gains. The model is evaluated on a comprehensive set of EO tasks. Across these tasks, the reported improvements are consistently positive and often sizable.

**Weaknesses:**

1. Misstated contribution. The paper positions “a novel dual-objective reward within GRPO” as its core novelty, but dual (format + accuracy) objectives are already established in R1/GRPO-style alignment. The true contribution here is the task-specific instantiation of accuracy rewards for EO tasks; the claims should be reframed accordingly.
2. Text-comparison rewards lack evidence of utility.
3. Non-verifiability & multi-validity: For a given scene, many captions can be valid. Lexical/semantic-overlap rewards primarily credit similarity to the reference, which can penalize reasonable paraphrases and under-penalize omissions.
4. Ablation signal is weak: Adding text-type rewards on top of SFT does not show clear gains in the reported ablations, leaving their practical benefit uncertain.
5. Comparisons omit thinking models. Most comparisons are against SFT models, while the paper itself uses an R1/GRPO recipe.-> Action: Add thinking-style baselines (e.g., the Qwen-VL-R1 used in Figure 9) for a fair comparison.

**Questions:**

1. Recall reward definition. How exactly is the recall reward computed and assigned? For single-label tasks, recall requires multiple samples—how is credit allocated per sample and across different rollouts? For multi-label tasks, the recall reward seems to be easy to be hacked (e.g., predicting all/most classes so get recall of 1)? Please provide more details of this reward.

2. The demos exhibit relatively short reasoning lengths, which seems inconsistent with the expected increase in reasoning length typically observed in R1-style training. Could you provide ablation results on the format reward in your setting to justify its necessity, given the absence of an increase in reasoning length?

---

### Official Review · Reviewer_24Qu · 2025-11-01

**Soundness:** 2
**Presentation:** 3
**Contribution:** 2
**Rating:** 2
**Confidence:** 4

**Summary:**

The paper proposes GeoVLM-R1, a reinforcement fine-tuning framework designed to enhance reasoning ability in remote sensing vision–language models. Built upon the GRPO (Group Relative Policy Optimization) algorithm, the approach introduces task-aware dual-objective rewards that jointly consider output format and semantic accuracy. The authors conduct experiments on multiple Earth Observation benchmarks, including classification, captioning, grounding, and change detection, and report consistent improvements over existing baselines. The paper also provides ablation studies and visual examples to analyze the effects of different reward components. Overall, it explores the potential of reinforcement learning–based reasoning in domain-specific multimodal settings.

**Strengths:**

- The work adapts GRPO-based reinforcement fine-tuning to Earth Observation tasks and integrates task-aware rewards, providing a modest yet meaningful extension to existing reasoning frameworks.

- Experiments across multiple benchmarks demonstrate consistent, though moderate, improvements over baseline models. The ablation studies effectively clarify the roles of different components.

- The paper is visually well presented—figures and tables are thoughtfully designed and help convey complex results clearly. For example, Figure 1 offers a concise and engaging overview of model performance across various EO tasks, while Figures 2–3 illustrate the training pipeline and reward mechanism in an intuitive way, making the approach easy to follow.

**Weaknesses:**

The paper mainly introduces a post-training strategy based on GRPO for remote sensing MLLMs. In the Introduction, the authors state that the core of this strategy lies in GRPO’s dual-objective reward design: (i) accuracy compliance and (ii) format compliance. However, these aspects have already been extensively explored in existing GRPO-related works. Ultimately, the authors only add a task-aware accuracy reward mechanism.

I did not see sufficient analysis or discussion. For example:

1. Why is the dual-objective reward design of GRPO ineffective in remote sensing scenarios? What exactly makes the existing GRPO unsuitable for remote sensing-related tasks?
2. The task-aware accuracy reward introduces a separate reward for each task. First, this approach lacks generalization — enumerating tasks is not elegant. It would be better to consider a unified reward that captures the common characteristics of the entire remote sensing domain while maintaining generalization through a standardized framework. Second, I did not see an in-depth discussion of these rewards. Why are they effective? We expect a deeper, mechanism-level analysis and experiments.

Overall, I think this work presents a number of solid experiments, but both the method design and experimental analysis remain rather superficial.

**Questions:**

Below are some specific opinions.

(1) The paper claims that reinforcement learning (RL) enables deep reasoning, whereas supervised fine-tuning (SFT) leads to shallow reasoning. However, the authors compare an RL model trained on reasoning data with an SFT model that lacks such data, omitting a crucial baseline: an SFT model trained on high-quality Chain-of-Thought (CoT) data. This omission makes the comparison unfair and methodologically weak, as a strong CoT-SFT baseline might achieve similar or even better reasoning performance without the complexity and instability of RL. To substantiate the claim, the authors should include and analyze a CoT-SFT baseline to isolate the true contribution of RL to reasoning. Could the authors include a stronger SFT-on-CoT baseline to better isolate the specific contribution of reinforcement learning? This would help clarify whether the reported improvements genuinely result from the RL procedure rather than exposure to reasoning-style data.

(2) The paper claims to enhance reasoning ability, yet the evaluation relies solely on task-level metrics (e.g., IoU, F1, BLEU, CIDEr) that do not assess the quality or correctness of reasoning chains. This mismatch raises concern that improvements may result from superficial or templated reasoning traces rather than genuine reasoning enhancement. Moreover, while the paper introduces a “dual-objective reward” (format compliance + task semantics), the inclusion of a format compliance objective may further amplify templated CoT behavior—encouraging the model to produce reasoning traces that look structured but lack genuine logical depth. The paper lacks analysis of:
Reward-hacking behavior — cases where the model generates format-correct but semantically incorrect reasoning. The current evaluation primarily measures task-level accuracy but does not directly assess the quality or consistency of the reasoning traces. How might the authors incorporate a more explicit evaluation of reasoning quality to strengthen the empirical claims?

(3) We find that the paper’s task-aware reward design, though novel in concept, fundamentally limits scalability and generality. The authors design separate, differentiable rewards for each downstream task, requiring manual engineering of task-specific signals. This approach contradicts the goal of building a general-purpose or “foundation” model and becomes impractical for large-scale or open-domain settings.  Does the task-aware reward design scale to new tasks without manual redesign? If not, could a more unified or adaptive reward formulation improve generality?

---

### Official Review · Reviewer_8WsC · 2025-11-01

**Soundness:** 3
**Presentation:** 3
**Contribution:** 3
**Rating:** 4
**Confidence:** 5

**Summary:**

EO tasks introduce unique challenges, including referred object detection, image/region captioning, change detection, grounding, and temporal analysis, which demand task-aware reasoning. Most of RSVLM rely on supervised instruction tuning and text priors, so their “reasoning” capability is shallow. To address this, the paper proposes GeoVLM-R1, a post-training RL framework for reasoning capabilities in EO tasks, including two-stage post-training: (i) first, a supervised fine-tuning stage on diverse EO tasks from EarthDial to give the model broad geospatial competence; (ii) then, a GRPO-based reinforcement stage that uses format and correctness reward. Experiments are conducted on 28 downstream benchmarks, and show impressive performance.

**Strengths:**

1. The paper designs 9 task-aware reward functions. The rewards for captioning are novel yet straightforward. SBERT, Lexical-Metric-based Grounding, Levenshtein Similarity Ratio, Jaccard Similarity and Hybrid SBERT and Lexical-Metric rewards seems appropriate for their corresponding tasks.

2. The experimental improvements demonstrate that the proposed SFT + GRPO pipeline is effective for multiple downstream EO tasks.

3. The overall pipeline and training design are clearly presented, which makes the method relatively easy to reproduce.

**Weaknesses:**

1. The paper does not clearly state whether RL is run per task separately or with mixed-task batches, nor how rewards with different value ranges are normalized. This is important for reproducibility.

2. Tasks such as visual grounding in EO are not fully explored. It would strengthen the paper if the authors could report results on VG benchmarks such as VRSBench or RSVG-DIOR.

3. For the classification task, the GRPO-tuned model does not seem to outperform the SFT-only model by a large margin. Could the authors provide possible explanations for this observation (see Fig. 5)?

4. The paper does not appear to discuss the problem of data/task imbalance. Do the authors use any strategies such as re-sampling or task-level weighting to alleviate the imbalance across different tasks?

5. Could the authors explain the intuition for designing 3 different captioning rewards?

6. In Fig. 7, the performance differences among different rewards are not very significat. Can the authors comment on why this is the case?

The score will be changed if authors can address some of concerns.

**Questions:**

See weaknesses.

---

### Official Review · Reviewer_pPH2 · 2025-11-03

**Soundness:** 2
**Presentation:** 2
**Contribution:** 3
**Rating:** 4
**Confidence:** 4

**Summary:**

This paper introduces GeoVLM-R1, which the authors claim is “a RL framework designed to enhance structured reasoning for complex EO tasks.” The “framework” comprises two stages: (1) supervised fine-tuning (SFT) on reasoning traces for EO tasks, and (2) RL training using group relative policy optimization (GRPO). In experiments, the paper finetunes a Qwen2.5VL-3B-Instruct VLM on the EarthDial-Instruct dataset, which is a remote sensing instruction-following dataset with 10 tasks.

The key contributions of this paper are:

1. Proposing to combine SFT with GRPO to train an EO-VLM.

2. Designing hand-crafted RL reward functions for each downstream task to use in GRPO training.

The authors find that on the EarthDial-Instruct dataset, the proposed SFT+RL training paradigm with the Qwen2.5VL-3B-Instruct VLM outperforms baselines.

**Strengths:**

**S1) Strong performance compared to baselines**

The authors demonstrate improved performance compared to baselines (especially the EarthDial VLM) across the board on the EarthDial-Instruct dataset.

**S2) Extensive evaluations and ablation studies**

The ablation studies show that combination of SFT + GRPO with specific choices of RL reward functions leads to significant gains in performance.

**Weaknesses:**

**W1) Reproducibility concerns**

My primary concern with the paper is reproducibility of results. First, there is no code provided, whether as a link to an anonymous code repo, or as a supplementary ZIP. Second, the paper does not sufficiently discuss the data/tasks used and the metrics computed. These details do not need to be in the main paper, but they should at least be included in an appendix. See W4) for more questions about metrics.

**W2) The “novelty” of the GeoVLM-R1 framework is not clearly explained**

My understanding is that the “GeoVLM-R1” framework refers to applying the combination of SFT + GRPO to earth observation data. Using SFT+GRPO in general is not new for LLMs, so is the novelty its application to earth observation data? If so, this should be clearly stated to avoid any misinterpretation. Currently, the following statement on lines 134-136 may be easily misinterpreted to suggest that the paper is the first to propose this combination.

> “We propose GeoVLM-R1, a RL framework designed to enhance structured reasoning for complex EO tasks. Our method adopts a two-stage training paradigm (Fig. 2), combining supervised finetuning (SFT) with R1-style post-training based on GRPO (Shao et al., 2024).”

**W3) Unclear whether GeoVLM-R1 “framework” is actually superior to non-RL training**

The authors claim that the GeoVLM-R1 “framework” (SFT+GRPO on EO data) outperforms existing baselines. The authors fine-tune a Qwen2.5VL-3B-Instruct base model using the GeoVLM-R1 framework. Henceforth, I will refer to this model as “Qwen2.5VL/GeoVLM-R1.”

In contrast, the main point of comparison is the EarthDial VLM, which pairs an InternViT-300M visual encoder with a Phi-3-mini LLM; EarthDial is not trained with any RL objective.

While it’s true that the overall performance metrics of Qwen2.5VL/GeoVLM-R1 are higher than EarthDial, it’s unclear whether the improvements are due to the GeoVLM-R1 training framework, or due to the model architecture. Unfortunately, the ablation study presented in Section 4.3 does not address this key question.

To address whether the performance gains come from model architecture or training methodology, the authors should design ablation studies to address this specific question.

**W4) Paper writing lacks some clarity and details**

Overall, the paper lacks clarity and details across both the methodology and experimental setup.

- The metrics shown in each table should be explained, at least in the appendix if there is no space in the main paper.

- Please provide more details on how Sentence-BERT is used.

- In lines 250/251, why are negative rewards not allowed by your framework?

- Citations for the different reward metrics should be given, unless the metrics are truly novel.

- What is the “Baseline” method in Figure 8?


Minor formatting mistakes (did not affect score):

- Table captions should be above the tables, not below the tables.

- Figure 2: it’s unclear what the input to the model is that causes the RL Model Output.

**Questions:**

Q1) Does the GeoVLM-R1 handle multispectral EO inputs? If so, what are the architectural changes to the Qwen2.5VL base model needed to accommodate multispectral images?

Q2) Could you please explain the Rouge1, Rouge-L, and Meteor metrics? No citation for these metrics is given.

---

### Note · Authors · 2025-11-12

I have read and agree with the venue's withdrawal policy on behalf of myself and my co-authors.